# Polygenic prediction via Bayesian regression and continuous shrinkage priors

Tian Ge[1,2,3], Chia-Yen Chen [1,2,3,4], Yang Ni [5], Yen-Chen Anne Feng[1,2,3,4] & Jordan W. Smoller[1,2,3]

Polygenic risk scores (PRS) have shown promise in predicting human complex traits and diseases. Here, we present PRS-CS, a polygenic prediction method that infers posterior effect sizes of single nucleotide polymorphisms (SNPs) using genome-wide association summary statistics and an external linkage disequilibrium (LD) reference panel. PRS-CS utilizes a high-dimensional Bayesian regression framework, and is distinct from previous work by placing a continuous shrinkage (CS) prior on SNP effect sizes, which is robust to varying genetic architectures, provides substantial computational advantages, and enables multivariate modeling of local LD patterns. Simulation studies using data from the UK Biobank show that PRS-CS outperforms existing methods across a wide range of genetic architectures, especially when the training sample size is large. We apply PRS-CS to predict six common complex diseases and six quantitative traits in the Partners HealthCare Biobank, and further demonstrate the improvement of PRS-CS in prediction accuracy over alternative methods.

[1] Psychiatric and Neurodevelopmental Genetics Unit, Center for Genomic Medicine, Massachusetts General Hospital, Boston, MA 02114, USA. [2] Department of Psychiatry, Massachusetts General Hospital, Harvard Medical School, Boston, MA 02114, USA. [3] Stanley Center for Psychiatric Research, Broad Institute of MIT and Harvard, Cambridge, MA 02142, USA. [4] Analytic and Translational Genetics Unit, Center for Genomic Medicine, Massachusetts General Hospital, Boston, MA 02114, USA. [5] Department of Statistics, Texas A&M University, College Station, TX 77843, USA. Correspondence and requests for materials should be addressed to T.G. (email: tge1@mgh.harvard.edu)

Polygenic risk scores (PRS), which summarize the effects of genome-wide genetic markers to measure the genetic liability to a trait or a disorder, have shown promise in predicting human complex traits and diseases, and may facilitate early detection, risk stratification, and prevention of common complex diseases in healthcare settings[1,2].

To maximize the translational potential of PRS, statistical and computational methods are needed that can (1) jointly model genetic markers across the genome to make full use of the available information while accounting for local linkage disequilibrium (LD) structures; (2) accommodate varying effect size distributions across complex traits and diseases, from highly polygenic genetic architectures (e.g., height and schizophrenia), to a mixture of small effect sizes and clusters of genetic loci that have moderate to larger magnitudes of effects (e.g., autoimmune diseases and Alzheimer's disease); (3) produce prediction from summary statistics of genome-wide association studies (GWAS) without access to individual-level data; and (4) retain computational scalability.

To date, most applications calculate PRS from a subset of the genetic markers after pruning out single nucleotide polymorphisms (SNPs) in LD and applying a P-value threshold to GWAS summary statistics[3]. Although this approach has advantages in terms of computational and conceptual simplicity, and has been used to predict genetic liability across a broad phenotypic spectrum, recent studies have shown that this conventional method for PRS construction discards information and limits prediction accuracy[4]. More sophisticated Bayesian polygenic prediction methods that rely on GWAS summary statistics, including LDpred[4] and the normal-mixture model recently developed[5,6], can incorporate genome-wide markers and accommodate varying genetic architectures, and thus have enhanced performance and flexibility. However, the type of prior on SNP effect sizes used in these methods, known as discrete mixture priors, imposes daunting computational challenges and may result in inaccurate adjustment for local LD patterns.

In this work, we present a polygenic prediction method, PRS-CS, which utilizes a Bayesian regression framework and places a conceptually different class of priors—the continuous shrinkage (CS) priors—on SNP effect sizes. Continuous shrinkage priors allow for marker-specific adaptive shrinkage (i.e., the amount of shrinkage applied to each genetic marker is adaptive to the strength of its association signal in GWAS), and thus can accommodate diverse underlying genetic architectures. In addition, continuous shrinkage priors enable conjugate block update of the SNP effect sizes in posterior inference (i.e., effect sizes for SNPs in each LD block are updated jointly, in a multivariate fashion, in contrast to updating the effect size for each marker separately and sequentially), and thus can accurately model local LD patterns and provide substantial computational improvements. Several special cases of continuous shrinkage priors have been applied to quantitative trait prediction or gene mapping[7–12]. However, all previous work required individual-level data and was limited to small-scale analyses (both in term of the sample size and number of genetic markers). PRS-CS only requires GWAS summary statistics and an external LD reference panel, and therefore can be applied in a broader range of settings.

We conduct simulation studies using the UK Biobank genetic data[13,14], and demonstrate that PRS-CS dramatically improves the predictive performance of PRS over existing methods across a wide range of genetic architectures, especially when the training sample size is large. We apply PRS-CS to predict six curated common complex diseases (breast cancer (BRCA), coronary artery disease (CAD), depression (DEP), inflammatory bowel disease (IBD), rheumatoid arthritis (RA), and type 2 diabetes mellitus (T2DM)) and six quantitative traits (height, body mass index, high-density lipoproteins, low-density lipoproteins, cholesterol, and triglycerides) in the Partners HealthCare Biobank[15], and further demonstrate the potential of PRS-CS for the clinical translation of polygenic prediction.

## Results

**Conceptual frameworks**. We consider a Bayesian high-dimensional regression framework for polygenic modeling and prediction:

$$\mathbf{y}_{N \times 1} = \mathbf{X}_{N \times M}\boldsymbol{\beta}_{M \times 1} + \boldsymbol{\varepsilon}_{N \times 1}, \tag{1}$$

where $N$ and $M$ denote the sample size and number of genetic markers, respectively, $\mathbf{y}$ is a vector of traits, $\mathbf{X}$ is the genotype matrix, $\boldsymbol{\beta}$ is a vector of effect sizes for the genetic markers, and $\boldsymbol{\varepsilon}$ is a vector of residuals. By assigning appropriate priors on the regression coefficients $\boldsymbol{\beta}$ to impose regularization, additive PRS can be calculated using posterior mean effect sizes.

Essentially all widely used prior densities for $\boldsymbol{\beta}$ can be represented as scale mixtures of normals:

$$p(\beta_j) = \int N(0, \Psi_j)\mathrm{d}G(\Psi_j), \qquad j = 1, 2, \cdots, M, \tag{2}$$

or equivalently, as the following hierarchical form:

$$\beta_j | \Psi_j \sim N(0, \Psi_j), \qquad \Psi_j \sim G, \qquad j = 1, 2, \cdots, M, \tag{3}$$

where $N(\mu, \sigma^2)$ is a normal distribution with mean $\mu$ and variance $\sigma^2$, and $G$ is a mixing distribution. For example, if $G$ places all its mass at a single point, i.e., $G(\Psi_j) = \delta_{\sigma_\beta^2}$, where $\delta_{\bullet}$ is the Dirac delta measure, then marginally $\beta_j \sim N(0, \sigma_\beta^2)$, and we have recovered the infinitesimal model[16]. To create a more flexible model of the genetic architecture, a discrete mixture of two or more point masses or densities can be used, which allows for a wider effect size distribution than a normal prior can produce. For example, $G(\Psi_j) = (1 - \pi)\delta_0 + \pi\delta_{\tau^2}$, where $\pi$ is the mixing probability (the fraction of causal variants), produces the point-normal prior on effect sizes, $\beta_j \sim (1-\pi)\delta_0 + \pi N(0, \tau^2)$, which was used in LDpred[4]. Although discrete mixture priors offer a natural and intuitive approach to model non-infinitesimal genetic architectures, posterior inference requires a stochastic search over an exponentially large discrete model space, and does not allow for multivariate block update of effect sizes, which limits computational efficiency and may result in inaccurate modeling of local LD patterns.

In this work, we investigate a conceptually different class of priors—the continuous shrinkage priors. In particular, we consider the following prior on SNP effect sizes, which can be represented as global-local scale mixtures of normals:

$$\beta_j | \psi_j \sim N(0, \phi\psi_j), \qquad \psi_j \sim g, \tag{4}$$

where $\phi$ is a global scaling parameter that shares across genetic markers and controls the degree of sparseness of the model, and $g$ is an absolutely continuous density function, in contrast to a discrete mixture of atoms or densities. By appropriately choosing the continuous mixing density $g$, this modeling framework can produce a variety of shapes of the prior distribution on $\beta_j$. In particular, $g$ can be designed to introduce a prior distribution on the SNP effect sizes that has a sizable amount of mass near zero to impose strong shrinkage on noise, while at the same time has heavy tails to avoid over-shrinkage of truly non-zero effects. The marker-specific local shrinkage parameter $\psi_j$ can then adaptively squelch small noisy estimates towards zero, while leaving data-supported large signals unshrunk. In this work, we investigate a specific $g$ (known as the Strawderman-Berger prior[17,18]; see Methods section), and present two versions of the algorithm, which differ in the way to learn the global scaling parameter $\phi$. In

PRS-CS, we search a small number of fixed $\phi$, select the $\phi$ value that produces the best predictive performance in a validation data set, and evaluate the algorithm in an independent testing set. In the second version of the algorithm, which we call PRS-CS-auto, we use a fully Bayesian approach and place a standard half-Cauchy prior on the global shrinkage parameter[19,20]: $\phi^{1/2} \sim C^+(0, 1)$, such that $\phi$ is automatically learnt from data and no validation data set is needed.

Individual-level Bayesian regression models (1) with a prior on SNP effect sizes can often be approximated using an external LD reference panel and turned into summary statistics based methods[4,6,21,22]. Here we enable posterior inference of SNP effect sizes from GWAS summary statistics under continuous shrinkage priors using an efficient Gibbs sampler with multivariate block update of the effect sizes (see Methods section).

**Overview of polygenic prediction methods**. We compare PRS-CS and PRS-CS-auto with four polygenic prediction methods that rely on GWAS summary statistics in both simulations and real data analyses: polygenic scoring based on all genetic markers (unadjusted PRS), informed LD-pruning (also known as LD-clumping) and $P$-value thresholding (P+T), LDpred and LDpred-inf[4]. Throughout the paper, we use the 1000 Genomes Project (1 KG) European sample ($N = 503$) as the external LD reference panel, but also assess the impact of using an in-sample LD reference panel on prediction accuracy in Supplementary Information.

**Simulations**. We first compared the predictive performance of six polygenic prediction methods across different genetic architectures and training sample sizes (i.e., GWAS sample sizes) in simulation studies (Fig. 1 and Supplementary Table 1). SNP effect sizes were simulated using (1) a point-normal model with different numbers of causal variants, and (2) a normal mixture model, as described in the Methods section. Tuning parameters ($P$-value threshold in P+T, fraction of causal SNPs in LDpred, and global shrinkage parameter in PRS-CS) were selected in a validation data set ($N = 3000$). Prediction accuracy for all methods was quantified by $R^2$ between the observed and predicted traits in an independent testing set ($N = 3000$).

Figure 1 shows that polygenic prediction methods that do not account for non-infinitesimal genetic architectures (unadjusted PRS and LDpred-inf) performed poorly when the number of causal variants is small, but became more comparable to other methods when the genetic architectures are highly polygenic. For all the methods, the prediction accuracy decreased as the number of causal variants increases with fixed heritability, because as more causal SNPs are in LD (as a result of more causal SNPs being randomly sampled across the genome) and their effect sizes decline, it becomes increasingly difficult to distinguish real signals from noise. Overall, methods that account for local LD patterns (LDpred, PRS-CS, and PRS-CS-auto) outperformed P+T, which discards LD information. However, one unexpected observation is that, when the genetic architecture is sparse, the prediction accuracy of LDpred decreased dramatically as the training sample size grows. This is likely because when the number of causal variants is small and the training sample size is large, all markers in LD with the causal variant become highly statistically significant in association tests, and LDpred does not accurately adjust for the LD structure, resulting in a decrease in predictive performance. In contrast, PRS-CS and PRS-CS-auto were

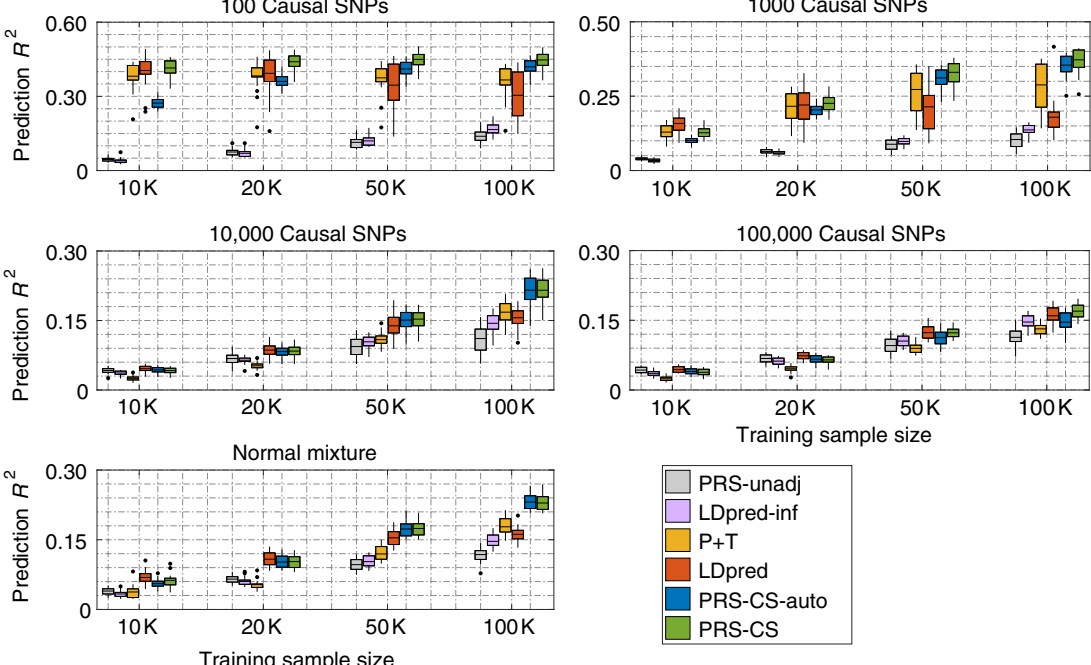

**Fig. 1** Predictive performance of six polygenic prediction methods in simulation studies using a point-normal model and a normal mixture model. Heritability was fixed at 0.5. The 1000 Genomes Project European sample was used as an external linkage disequilibrium (LD) reference panel. Tuning parameters ($P$-value threshold in P+T, fraction of causal markers in LDpred, and global shrinkage parameter in PRS-CS) were selected in a validation data set. Prediction accuracy was quantified by $R^2$ between the observed and predicted traits in an independent testing set. The upper four panels correspond to the four genetic architectures (100, 1000, 10,000, and 100,000 causal variants) simulated using the point-normal model. The lower panel corresponds to the normal mixture model. Within each panel, results for four different training sample sizes (10,000, 20,000, 50,000, and 100,000) are shown. On each box, the central mark is the mean across 20 simulations, the edges of the box are the 25th and 75th percentiles, the whiskers extend to the most extreme data points that are not considered outliers, and the outliers are plotted individually

minimally affected in the combination of sparse genetic architectures and large training sample sizes, which demonstrates the advantage of multivariate modeling and block update of the effect sizes for genetic markers in LD. In a few scenarios where the training sample size is small, PRS-CS produced lower prediction accuracy than LDpred, but it outperformed LDpred as the sample size grows across all genetic architectures. PRS-CS-auto did not perform well when the training sample size is small and the genetic architecture is sparse (e.g., in the case of 100 causal variants and 10,000 training samples), but approached the performance of PRS-CS as the sample size increases.

In addition to prediction accuracy, we assessed the calibration of polygenic prediction methods by regressing the true phenotype onto the PRS predictor and inspecting the regression slope. A slope close to one indicates that a predictor is correctly calibrated. Consistent with predictive performance, as the training sample size grows, our Bayesian approach provides the best calibration among all methods examined (Supplementary Table 7). PRS-CS-auto is particularly well calibrated for large training sample sizes, because it automatically learns the sparseness of the genetic architecture from data and adjusts for the LD structure accordingly.

Secondary simulation studies using (1) the point-normal model with different total heritability (0.2 and 0.8); (2) a point-$t$ model with different numbers of causal variants; and (3) a point-gamma model with different numbers of causal variants produced similar patterns of prediction accuracy (Supplementary Figs. 1–4; Supplementary Tables 2–5) and calibration properties (Supplementary Tables 8–11). Using the combined UK Biobank validation and testing data sets ($N = 6000$) as an in-sample LD reference panel in the point-normal simulations produced, in general, slightly higher prediction accuracy for methods making use of LD information (Supplementary Fig. 5; Supplementary Tables 6 and 12), suggesting that using a larger reference panel that better aligns with the LD structure of the target sample may increase predictive performance. However, as the improvement was marginal, it appears that the performance of PRS-CS(-auto) is not particularly sensitive to the LD reference panel, and

1KG can serve as a valid reference despite its relatively small sample size.

**Polygenic prediction in the Partners Biobank.** We applied PRS-CS, PRS-CS-auto, and alternative methods to predict six curated common complex diseases (breast cancer, coronary artery disease, depression, inflammatory bowel disease, rheumatoid arthritis, and type 2 diabetes mellitus), and six quantitative traits (height, body mass index, high-density lipoproteins, low-density lipoproteins, cholesterol, and triglycerides) in the Partners Health-Care Biobank. Large-scale GWAS summary statistics for each disease and trait were downloaded from public domains (Table 1 and Supplementary Data 1). SNP heritability for each disease (both on the observed scale and the liability scale) and trait estimated using GWAS summary statistics and LD score regression[23] are presented in Supplementary Table 13.

Predictive performance measured by Nagelkerke's $R^2$ (for disease phenotypes) and $R^2$ (for quantitative traits) is summarized in Fig. 2. Additional prediction accuracy metrics, including area under the receiver operating characteristic (ROC) curve (known as AUC), area under the precision-call curve, and the odds ratio (OR) comparing top 10% of the participants having high polygenic risk with the remaining 90% of the sample, produced similar results in terms of the ranked performance of polygenic prediction methods and are reported in Supplementary Data 2.

Consistent with previous work, unadjusted PRS performed poorly regardless of the genetic architecture, and LDpred showed an overall improvement over P+T. Among the six curated disease phenotypes, PRS-CS produced substantially better predictions for breast cancer (41.85% relative increase in Nagelkerke's $R^2$ compared to LDpred) and rheumatoid arthritis (28.62% relative increase in Nagelkerke's $R^2$ compared to LDpred). For coronary artery disease, depression and type 2 diabetes mellitus, LDpred and PRS-CS had similar predictive performance, and both performed dramatically better than P+T. PRS-CS was only inferior to LDpred in the prediction of inflammatory bowel disease (10.24% relative decrease in Nagelkerke's $R^2$). However,

**Table 1 Information on six common complex diseases and six quantitative traits**

| Disease/Trait | Abbreviation | GWAS reference | GWAS sample size (case/control) | 1KG ∩ PBK SNPs | 1KG ∩ PBK ∩ HM3 SNPs | PBK sample size (case/control) |
|---|---|---|---|---|---|---|
| Breast cancer | BRCA | Michailidou et al.[59] | 228,951 (122,977/105,974) | 5,022,127 | 857,616 | 10,220 (884/9336) |
| Coronary artery disease | CAD | Nikpay et al.[60] | 184,305 (60,801/123,504) | 4,803,592 | 849,399 | 16,251 (2759/13,492) |
| Depression | DEP | Wray et al.[61] | 173,005 (59,851/113,154) | 4,924,025 | 850,291 | 15,276 (2361/12,915) |
| Inflammatory bowel disease | IBD | Liu et al.[62] | 34,652 (12,882/21,770) | 4,823,570 | 849,749 | 18,998 (750/18,248) |
| Rheumatoid arthritis | RA | Okada et al.[63] | 58,284 (14,361/43,923) | 3,872,637 | 849,680 | 18,170 (753/17,417) |
| Type 2 diabetes mellitus | T2DM | Scott et al.[64] | 159,208 (26,676/132,532) | 4,901,848 | 856,912 | 18,823 (1978/16,845) |
| Height | HGT | Yengo et al.[65] | 693,529 | 1,578,533 | 750,888 | 3957 |
| Body mass index | BMI | Yengo et al.[65] | 681,275 | 1,579,905 | 751,676 | 3954 |
| High-density lipoproteins | HDL | Willer et al.[66] | 188,578 | 1,604,577 | 758,036 | 2491 |
| Low-density lipoproteins | LDL | Willer et al.[66] | 188,578 | 1,600,625 | 756,724 | 1713 |
| Cholesterol | CHOL | Willer et al.[66] | 188,578 | 1,604,391 | 757,970 | 2561 |
| Triglycerides | TRIG | Willer et al.[66] | 188,578 | 1,601,270 | 756,913 | 2505 |

The sample size for each external genome-wide association study (GWAS), and the number of genetic markers included in the polygenic prediction are shown, along with the sample size for each disease and quantitative phenotype in the Partners HealthCare Biobank (PBK). For unadjusted PRS and P+T, all common genetic markers (minor allele frequency ≥1%) that passed quality control and are present in the summary statistics and 1000 Genomes Project (1KG) European sample were used in prediction. For LDpred(-inf) and PRS-CS(-auto), genetic markers were further restricted to the HapMap3 (HM3) panel

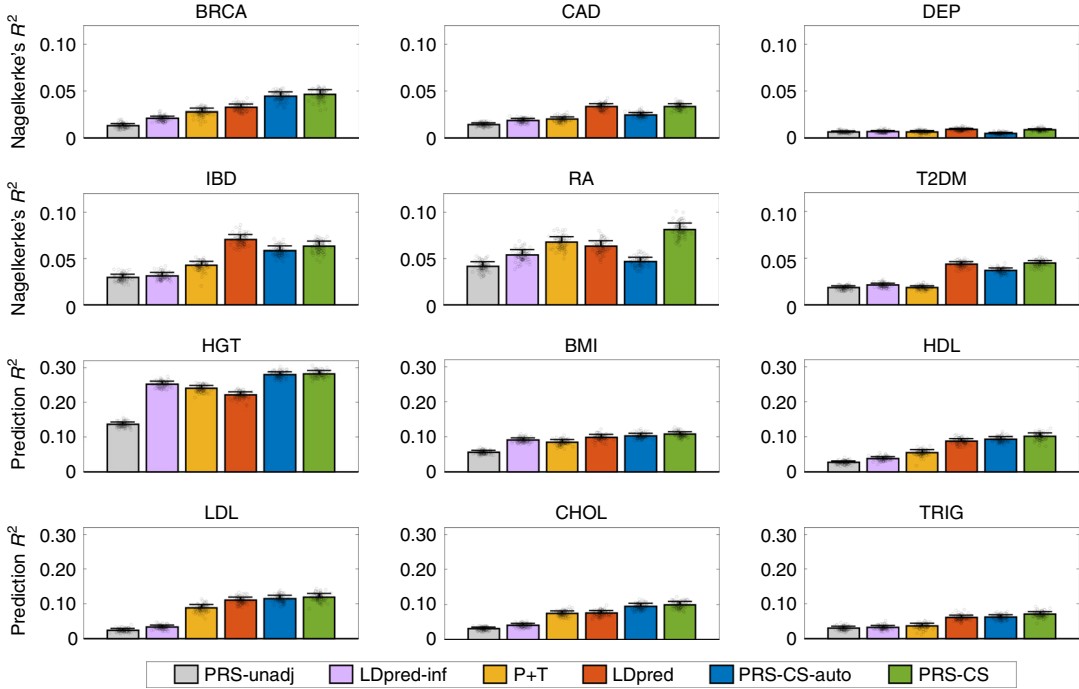

**Fig. 2** Prediction accuracy of six polygenic prediction methods in the Partners HealthCare Biobank. Posterior effect sizes of single nucleotide polymorphisms (SNPs) were trained with large-scale genome-wide association summary statistics, using the 1000 Genomes Project European sample as an external linkage disequilibrium (LD) reference panel. Polygenic scores were applied to predict six curated common complex diseases—breast cancer (BRCA), coronary artery disease (CAD), depression (DEP), inflammatory bowel disease (IBD), rheumatoid arthritis (RA), and type 2 diabetes mellitus (T2DM), and six quantitative traits—height (HGT), body mass index (BMI), high-density lipoproteins (HDL), low-density lipoproteins (LDL), cholesterol (CHOL), and triglycerides (TRIG). The Partners HealthCare Biobank sample for each disease and quantitative phenotype was repeatedly and randomly split into a validation set comprising 1/3 of the data and a testing set comprising 2/3 of the data. Tuning parameters (*P*-value threshold in P+T, fraction of causal SNPs in LDpred, and global shrinkage parameter in PRS-CS) were selected in the validation data set, and the predictive performance was assessed in the testing set. For disease (case–control) phenotypes and quantitive traits, prediction accuracy was measured by the Nagelkerke's $R^2$ and $R^2$, respectively, averaged across 100 random splits. The error bar indicates the standard deviation of prediction accuracy across 100 random splits. Prediction accuracy for each random split is overlaid on the bar plot (black circles)

we note that inflammatory bowel disease has the smallest training sample size among all diseases and traits (Table 1). The lower prediction accuracy of PRS-CS for this disease is thus consistent with our simulation studies, where we observed that when the training sample size is limited, LDpred can outperform PRS-CS. PRS-CS-auto produced lower prediction accuracy than LDpred except for breast cancer, indicating that the current GWAS sample sizes for most diseases may not be large enough to accurately learn the global shrinkage parameter from GWAS summary statistics.

For the six quantitative traits, both PRS-CS and PRS-CS-auto consistently outperformed all alternative methods examined. The relative improvement in prediction accuracy for PRS-CS compared to LDpred ranged from 8.01% for LDL and 8.75% for BMI, to 27.75% for height and 32.05% for cholesterol, with an average improvement of 18.17%. The average improvement of PRS-CS-auto relative to LDpred across the six quantitative traits was 11.41%. The average improvements of PRS-CS and PRS-CS-auto relative to P+T were 48.16% and 38.62%, respectively. We note that LDpred was the best method after PRS-CS and PRS-CS-auto for all quantitative traits except height, for which its prediction accuracy was lower than LDpred-inf and P+T. This is theoretically expected and consistent with a recent study, which also observed that for highly polygenic traits, LDpred-inf often outperforms LDpred[24].

Overall, using the Partners HealthCare Biobank data as an in-sample LD reference (N = 19,136) instead of the 1KG reference

panel slightly increased the prediction accuracy but the improvement was marginal (Supplementary Fig. 6 and Supplementary Data 3).

## Discussion

Polygenic prediction, which exploits genome-wide genetic markers to estimate the genetic liability to a complex human disease or trait, is likely to become useful in clinical care and contribute to personalized medicine. As a high-dimensional regression problem that requires regularization, a majority of the existing methods that jointly model genetic markers across the genome employ Bayesian approaches and assign a discrete mixture prior on SNP effect sizes. Although intuitively appealing, this class of priors generates daunting computational challenges: the model space grows exponentially with the number of markers, which is difficult to fully explore, and more importantly, discrete mixture priors do not allow for block update of effect sizes and thus hinder accurate LD adjustment in polygenic prediction. LDpred[4] partially addressed this issue by making several simplifying assumptions to the posterior distribution and using marginal posterior without LD to approximate the true posterior. However, our simulation studies suggest that this approximation may be inaccurate.

We have presented a conceptually different class of priors—the continuous shrinkage priors—which can be represented as global-local scale mixtures of normals, for polygenic modeling. By using

a continuous mixing density on the scales of the marker effects, continuous shrinkage priors enable a simple and efficient Gibbs sampler with multivariate block update of the effect sizes, and thus resolve a major technical hurdle of discrete mixture priors. A second feature of the continuous shrinkage prior is its ability to shrink adaptively. By constructing a prior density on SNP effect sizes that is both peaked at zero and heavy-tailed, the method imposes strong shrinkage on small effects that are likely to be noise, while applying practically no shrinkage to data-supported truly non-zero signals. Simulated and real data analyses showed that PRS-CS consistently outperforms existing methods across a wide range of genetic architectures, especially when the training sample size is large. We note that previous work often extrapolated prediction accuracy for larger effective sample sizes by restricting the analysis to a subset of the genetic markers[4,24]. However, our simulations suggest that this approach may not fully capture the behavior of a polygenic prediction algorithm when the training sample size grows, and underscore the need for actually scaling up the sample size in future studies.

PRS-CS has a tuning parameter, i.e., the global shrinkage parameter $\phi$, which needs to be fixed based on prior beliefs about the sparseness of the genetic architecture, or selected by testing a small number of values. If a grid search is used, like other polygenic prediction methods that have tuning parameters such as P +T and LDpred, the optimal value of $\phi$ should be selected using a validation data set that is independent of the testing set where predictive performance is assessed to avoid overfitting. In this work, we also presented PRS-CS-auto, a fully Bayesian approach that enables automatic learning of $\phi$ from GWAS summary statistics. Although analyses in the Partners Biobank indicate that, for many disease phenotypes, the current GWAS sample sizes may not be large enough to accurately learn $\phi$ and the prediction accuracy of PRS-CS-auto may be lower than PRS-CS and LDpred, simulation studies and quantitative trait analyses suggest that PRS-CS-auto can be useful when the training sample size is large or when an independent validation set is difficult to acquire.

Although continuous shrinkage priors enable multivariate modeling of the LD structure, simultaneous updating of the effect sizes for genome-wide markers remains computationally infeasible. In this work, we used a genome partition computed and validated by prior work[25], which divides the genome into 1703 largely independent genomic regions, and has been successfully used in local heritability and genetic correlation analyses[26,27]. Block update of posterior SNP effect sizes can thus be performed within each LD block, assuming no LD between blocks. Using a sliding window approach as implemented in LDpred[4] may capture LD across blocks more accurately, but is more memory intensive and computationally expensive. By restricting the analysis to HapMap3 variants, the partition we employed gives a moderate number of SNPs within each block (on average ~500 SNPs per block), and the Bayesian computation with 1000 MCMC iterations on the longest chromosome can be completed within an hour using one Intel(R) Xeon(R) CPU core and 2 GB of memory. Expanding the size of LD blocks may improve prediction accuracy but increases computational cost (as each MCMC iteration requires inverting an $L \times L$ matrix where $L$ is the block size), while reducing the size of LD blocks has the potential risk of missing long-range LD. Therefore, the partition we chose represents a balance between modeling accuracy and computational burden. Including multi-million SNP predictors may increase prediction accuracy[28] but requires further work.

We note that the prior we investigated in this work, i.e., the Strawderman-Berger prior on the local marker-specific shrinkage parameter, is only one of the possible choices within the class of continuous shrinkage priors, which includes the normal-gamma prior[29,30], the normal-inverse-gaussian prior[29], the generalized $t$ (generalized double Pareto) prior[31,32], and the normal-exponential-gamma prior[33,34], among others. In addition, most frequentist regularization procedures, such as LASSO, elastic net and bridge regression, have a Bayesian counterpart that can be represented as global-local scale mixtures priors in combination with posterior mode inferences. Each of these priors uses a different continuous mixing density to produce a different marginal prior on the SNP effect sizes. These alternatives may perform equally well or better than the Strawderman-Berger prior for certain genetic architectures. However, we found that as long as the prior on the effect sizes places a sizable amount of mass around zero and has heavier-than-exponential tails, variation in the shape of the prior does not seem to have a large impact on prediction accuracy. Therefore, we believe that the primary gain of PRS-CS over existing methods lies in its more accurate multivariate modeling of local LD patterns and its block-updated Gibbs sampling that can improve the mixing and convergence rate of the Markov chain. We thus recommend using the Strawderman-Berger prior as a default choice. A systematic investigation and comparison of different continuous shrinkage priors is a direction of future work.

We note several additional directions for further technical developments that may be useful. First, although this paper is focused on polygenic prediction methods that only require GWAS summary statistics, PRS-CS, and PRS-CS-auto can be straightforwardly applied to individual-level data. Given that a majority of the existing Bayesian genomic prediction models, including Bayes alphabetic methods[10,35–40], BayesR[41,42], BVSR[43], BSLMM[44], and DPR[45], have used discrete mixture priors on SNP effect sizes, we expect that PRS-CS can provide substantial improvements in computational efficiency and prediction accuracy for genomic prediction that leverages individual-level data. Second, jointly modeling multiple genetically correlated traits and including functional annotations in polygenic modeling are expected to increase the predictive performance of PRS, as shown by recent studies[24,46,47]. Lastly, current research on polygenic prediction has largely been restricted to European samples. Heterogeneity between the GWAS, LD reference and testing samples may reduce prediction accuracy as recently demonstrated in genetic correlation analysis and fine-mapping[48,49]. Expanding genomic prediction methods to handle unknown ancestry of the target sample (e.g., applications in forensic science) and enable transethnic risk prediction is critical to maximize the value of PRS in a diverse population.

Although PRS-CS provides a substantial improvement over existing methods for polygenic prediction, current prediction accuracy of PRS is still lower than what can be considered clinically useful, and much work is needed to further improve the predictive performance and translational value of PRS. In theory, the utility of PRS depends on multiple factors, including the GWAS sample size, and the heritability and genetic architecture of the disease. For example, among the six complex diseases we analyzed, depression had the lowest prediction accuracy (Nagelkerke's $R^2$ less than 1%), likely due to a combination of its relatively low heritability, extremely polygenic genetic architecture, and the heterogeneous nature of the disorder. A recent study projected that a GWAS with multi-million subjects is needed to identify genetic variants that explain 80% of the SNP heritability for major depressive disorder[5]. In contrast, it may be easier to produce a clinically useful prediction for some autoimmune diseases or late-onset chronic diseases (e.g., coronary artery disease and type 2 diabetes), due to the existence of SNPs with moderate to larger effect sizes. With these being said, as the GWAS sample size continues to grow, we believe that the predictive value of PRS will keep increasing, and PRS-CS(-auto) will

demonstrate bigger advantages over existing methods with larger training sample sizes.

## Methods

**PRS-CS and PRS-CS-auto.** We consider the following phenotype model:

$$\mathbf{y} = \mathbf{Z}\boldsymbol{\beta} + \boldsymbol{\varepsilon}, \qquad \boldsymbol{\varepsilon} \sim N(0, \sigma^2 \mathbf{I}), \qquad p(\sigma^2) \propto \sigma^{-2}, \qquad (5)$$

where $\mathbf{y}$ is a vector of standardized phenotypes from $N$ individuals, $\mathbf{Z}$ is an $N \times M$ matrix of standardized genotypes (each column is mean centered and has unit variance), $\boldsymbol{\beta}$ is a vector of effect sizes, $\boldsymbol{\varepsilon}$ is a vector of independent environmental effects, and we have assigned a non-informative scale-invariant Jeffreys prior on the residual variance $\sigma^2$. In contrast to discrete mixture priors, we consider a conceptually different class of priors:

$$\beta_j \sim N\left(0, \frac{\sigma^2}{N}\phi\psi_j\right), \qquad \psi_j \sim g, \qquad (6)$$

where the variance of $\beta_j$ scales with the residual variance and the sample size, $\phi$ is a global scaling parameter that is shared across all effect sizes, $\psi_j$ is a local, marker-specific parameter, and $g$ is an absolutely continuous mixing density function. This type of prior is known as global-local scale mixtures of normals.

We first note that, given variance parameters $\sigma^2$, $\phi$ and $\psi_j$, $j = 1,2,\ldots, M$, and the marginal least squares effect size estimates of the regression coefficients $\hat{\boldsymbol{\beta}} = \mathbf{Z}^T\mathbf{y}/N$, the posterior mean of $\boldsymbol{\beta}$ is

$$E[\boldsymbol{\beta}|\hat{\boldsymbol{\beta}}] = (\mathbf{D} + \mathbf{T}^{-1})^{-1}\hat{\boldsymbol{\beta}}, \qquad (7)$$

where $\mathbf{T} = \text{diag}\{\phi\psi_1, \phi\psi_2, \ldots, \phi\psi_M\}$ is a diagonal matrix, and $\mathbf{D} = \mathbf{Z}^T\mathbf{Z}/N$ is the LD matrix. It can be seen that the posterior mean is a matrix shrinkage version of the least squares estimate. In the degenerative special case where $\psi_j \equiv 1$, the model becomes Ridge regression and all effect sizes are shrunk towards zero at the same constant rate controlled by the overall shrinkage parameter $\phi$. The introduction of the local shrinkage parameter $\psi_j$ thus allows heterogeneity in the scales of effect sizes.

To provide further intuitions, assuming that all genetic markers are unlinked (i.e., no LD), we have $\mathbf{D} = \mathbf{I}$ and thus

$$E[\beta_j|\hat{\beta}_j] = \frac{1}{1 + \phi^{-1}\psi_j^{-1}}\hat{\beta}_j = \left(1 - \frac{1}{1 + \phi\psi_j}\right)\hat{\beta}_j := (1 - \tau_j)\hat{\beta}_j, \qquad (8)$$

where $\tau_j = 1/(1 + \phi\psi_j)$ is the shrinkage factor for the $j$-th marker, which relies on both $\phi$ and $\psi_j$, and describes the amount of shrinkage from the marginal least squares solution towards zero; $\tau_j = 0$ indicates no shrinkage while $\tau_j = 1$ yields total shrinkage. Therefore, $\phi$ controls the overall sparsity level of the model and plays a similar role as the regularization parameter in penalized regression, while $\psi_j$ adaptively modifies the amount of shrinkage for each marker. By assigning a prior on $\psi_j$, which can produce a marginal prior density on $\beta_j$ that has both a sharp peak at zero and heavy tails, the model can pull small effects towards zero, while asserting little influence on larger effects.

In this work, we investigate a specific continuous shrinkage prior. We assign an independent gamma-gamma prior on the local shrinkage parameter $\psi_j$:

$$\psi_j \sim G(a, \delta_j), \qquad \delta_j \sim G(b, 1), \qquad (9)$$

where $G(\alpha, \beta)$ denotes the gamma distribution with shape parameter $\alpha$ and scale parameter $\beta$. By using change of variables, it can be verified that placing a gamma-gamma prior on $\psi_j$ is equivalent to placing a three-parameter beta (TPB) prior on the shrinkage factor $\tau_j$[33]:

$$\tau_j \sim \text{TPB}(a, b, \phi), \qquad (10)$$

where the TPB distribution has the following density function:

$$f(x; a, b, \phi) = \frac{\Gamma(a+b)}{\Gamma(a)\Gamma(b)}\phi^b x^{b-1}(1-x)^{a-1}\{1 + (\phi-1)x\}^{-(a+b)}, \qquad (11)$$

with $0 < x < 1$, $a > 0$, $b > 0$ and $\phi > 0$. When $\phi = 1$, the TPB distribution becomes a standard Beta distribution. For a fixed value of $\phi$, $a$ controls the behavior of the TPB prior near one, and thus the behavior of the prior on $\beta_j$ around zero; $b$ controls the behavior of the TPB prior near zero, and thus affects the tails of the prior on $\beta_j$. Figure 3 shows the prior densities on $\tau_j$ (upper panel) and $\beta_j$ (middle and lower panels) with $\phi = 1$, $b = 1/2$, and three different values of $a$: $a = 1/2$, $a = 1$ and $a = 3/2$. It can be seen that when $a = 1/2$ and $b = 1/2$, the TPB prior has substantial mass near zero and one (Fig. 3, upper panel), and thus the corresponding prior density on $\beta_j$ has a very sharp peak around the origin, with zero being a pole (singular point; Fig. 3, middle panel), along with heavy, Cauchy-like tails (Fig. 3, lower panel). This prior is known as the horseshoe prior[50], due to the horseshoe-shaped prior density on the shrinkage factor $\tau_j$. As $a$ increases, the prior on $\beta_j$ becomes less peaked at zero but the tails remain heavy. Finally, for fixed $a$ and $b$, decreasing the global shrinkage parameter $\phi$ shifts the TPB prior from left to right, which imposes stronger shrinkage on the regression coefficients $\beta_j$.

For all continuous shrinkage priors that take the general form in Eq. (6), Gibbs samplers with block update of the regression coefficients $\boldsymbol{\beta}$ (i.e., SNP effect sizes) can be easily derived. By using LD information from an external reference panel,

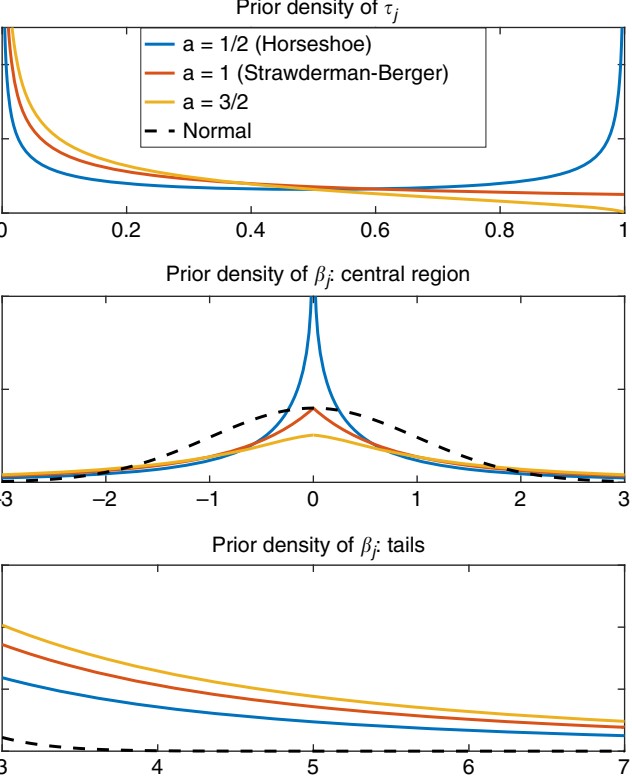

**Fig. 3** Densities of the priors. Upper panel: Density of the three-parameter beta prior on the shrinkage factor $\tau_j$ with $\phi = 1$, $b = 1/2$, and three different $a$ values. Middle panel: Central region of the marginal prior density on the effect size $\beta_j$ with $\phi = 1$, $b = 1/2$, and three different $a$ values, in comparison with the standard normal density. Lower panel: Tails of the marginal prior density on the effect size $\beta_j$ with $\phi = 1$, $b = 1/2$, and three different $a$ values, in comparison with the standard normal density

the method can be applied to GWAS summary statistics and does not require individual-level data. We describe the Gibbs sampler in Supplementary Note. In this study, we focus on a specific set of parameter values of the gamma-gamma prior on $\psi_j$ (or equivalently, the TPB prior on $\tau_j$): $a = 1$ and $b = 1/2$. This particular specification is known as the Strawderman-Berger prior[17,18] or the quasi-Cauchy prior[51], and appears to work well across a range of simulated and real genetic architectures.

In practice, we partition the genome into 1703 largely independent genomic regions estimated using data from the 1KG European sample[25–27] [http://bitbucket.org/nygcresearch/ldetect-data], and conduct multivariate update of the effect sizes within each LD block (see Supplementary Note). To avoid numerical issues caused by collinearity between SNPs, we set a lower bound on the amount of regularization applied to the genetic markers (i.e., restricting $\phi^{-1}\psi_j^{-1} \geq \rho$, where $\rho$ is a small constant). We use $\rho = 1$ throughout this paper.

We find that the predictive performance of the model is not sensitive to the global shrinkage parameter $\phi$, and setting $\phi^{1/2}$ roughly to the proportion of causal variants[52] works well. If a prior guess of the sparseness of the genetic architecture is not available, we provide two ways to learn $\phi$. In PRS-CS, we search a small number of $\phi$ values: $\phi^{1/2} \in \{0.0001, 0.001, 0.01, 0.1, 1\}$, and select the $\phi$ that produces the best predictive performance in a validation data set, which is independent of the testing set where prediction accuracy of the algorithm is evaluated. In PRS-auto, we use a fully Bayesian approach and assign a standard half-Cauchy prior on $\phi^{1/2}$[19,20], such that $\phi$ is automatically learnt from GWAS summary statistics and no validation data set is needed. See Supplementary Note for the Gibbs updates of $\phi$.

For both PRS-CS and PRS-CS-auto, the Gibbs sampler usually attains reasonable convergence after 1000 Markov Chain Monte Carlo (MCMC) iterations and produces prediction accuracy close to what can be achieved by much longer MCMC runs. We thus use 1000 MCMC iterations with the first 500 steps as burn-in in simulation studies to reduce computational cost. In practice, we recommend using longer MCMC runs when time and computational resources permit. In the Partners HealthCare Biobank analysis, we report the predictive performance of PRS-CS and PRS-CS-auto based on 10,000 MCMC iterations in total and 5000 burn-in steps.

**Unadjusted PRS**. The unadjusted PRS is the sum of all genetic markers across the genome, weighted by their marginal effect size estimates. More specifically, the unadjusted polygenic score for the $i$-th individual is $\text{PRS}_i = \sum_{j=1}^{M} X_{ij} \hat{b}_j$, where $M$ is the total number of genetic markers, $X_{ij}$ is the genotype for the $i$-th individual and the $j$-th SNP, and $\hat{b}_j$ is the estimated marginal per-allele effect size of the $j$-th SNP.

**P+T**. The P+T method refers to the calculation of PRS using informed LD-pruning (also known as LD-clumping) and $P$-value thresholding. In this study, we use the implementation of the P+T method in the software package PRSice-2[53] [https://choishingwan.github.io/PRSice] and its default parameter settings. Specifically, for any pair of SNPs that have a physical distance smaller than 250 kb and an $R^2$ greater than 0.1, the less significant SNP is removed. The polygenic score is then calculated as the sum of the remaining, largely independent SNPs with a GWAS association $P$-value below a threshold $P_T$, weighted by their marginal effect size estimates. We consider $P_T \in \{1E-8, 1E-7, 1E-6, 1E-5, 3E-5, 1E-4, 3E-4, 0.001, 0.003, 0.01, 0.03, 0.1, 0.3, 1\}$ in this paper. The $P_T$ value that produces the highest prediction accuracy in a validation data set is selected, and the predictive performance is assessed in an independent testing set.

**LDpred and LDpred-inf**. LDpred [https://github.com/bvilhjal/ldpred] is a method that infers the posterior mean effect size of each genetic marker from GWAS summary statistics while accounting for LD, using a point-normal prior on the SNP effect sizes and LD information from an external reference panel[4]. Consider the linear model $\mathbf{y} = \mathbf{Z}\boldsymbol{\beta} + \boldsymbol{\varepsilon}$, where both the phenotype $\mathbf{y}$ and the genotype matrix $\mathbf{Z}$ have been standardized. LDpred places an independent point-normal prior on each regression coefficient $\beta_j$:

$$\beta_j \sim \begin{cases} N\left(0, \frac{h_g^2}{\pi M}\right), & \text{with probability } \pi, \\ 0, & \text{with probability } 1 - \pi, \end{cases} \qquad (12)$$

where $h_g^2$ is the heritability explained by genome-wide genetic markers (known as SNP heritability), and $\pi$ is the fraction of causal variants. Given $\pi$ and an estimate of $h_g^2$, which can be obtained, for example, by applying LD score regression[23] to the GWAS summary statistics, LDpred employs an MCMC sampler to approximate the posterior mean of $\beta_j$, conditioning on marginal least squares effect size estimates and LD information from a reference panel. In this paper, we consider $\pi \in \{1E-5, 3E-5, 1E-4, 3E-4, 0.001, 0.003, 0.01, 0.03, 0.1, 0.3, 1\}$. The $\pi$ value with the highest prediction accuracy in a validation data set is selected, and the predictive performance is assessed in an independent testing set.

LDpred-inf is a special case of LDpred when all variants are assumed to be causal (i.e., $\pi = 1$). Under this infinitesimal model, the posterior mean effect sizes in the $\ell$-th LD window have a closed-form approximation:

$$\mathbb{E}[\boldsymbol{\beta}_\ell | \hat{\boldsymbol{\beta}}_\ell, \mathbf{D}_\ell] \approx \left(\mathbf{D}_\ell + \frac{M}{N h_g^2}\mathbf{I}\right)^{-1}\hat{\boldsymbol{\beta}}_\ell, \qquad (13)$$

where $\hat{\boldsymbol{\beta}}_\ell$ is a vector of marginal least squares effect size estimates, $\mathbf{D}_\ell$ is the LD matrix that can be estimated from an external reference panel, $\mathbf{I}$ is an identity matrix, and it has been assumed that $h_\ell^2$, the heritability explained by SNPs in the $\ell$-th LD window, is small such that $1 - h_\ell^2 \approx 1$. In this work, we use an LD radius of $M/3000$ to approximate local LD patterns, as suggested in Vilhjalmsson et al.[4]

**UK Biobank genetic data**. UK Biobank [http://www.ukbiobank.ac.uk] is a prospective cohort study of ~500,000 individuals recruited across Great Britain during 2006–2010[13]. The protocol and consent were approved by the UK Biobank's Research Ethics Committee. Data for the current analyses were obtained under an approved data request.

The genetic data for the UK Biobank comprises 488,377 samples and was phased and imputed to ~96 million variants with the Haplotype Reference Consortium (HRC) haplotype resource and the UK10K + 1KG reference panel. We leveraged the QC metrics provided by the UK Biobank[14] and removed samples that had mismatch between genetically inferred sex and self-reported sex, high genotype missingness or extreme heterozygosity, sex chromosome aneuploidy, and samples that were excluded from kinship inference and autosomal phasing. We further restricted the analysis to unrelated white British participants. We conducted simulation studies using 819,941 HapMap3 SNPs after removing ambiguous (A/T and C/G) SNPs and markers with minor allele frequency (MAF) <1%, missing rate >1%, imputation quality INFO score <0.8, and significant deviation from Hardy-Weinberg equilibrium (HWE) with $P < 1 \times 10^{-10}$. All genetic analyses in the UK Biobank were conducted using PLINK 1.9[54] [https://www.cog-genomics.org/plink/1.9].

**Simulations**. We performed simulation studies using real genetic data from the UK Biobank and the 1KG European sample ($N = 503$) as an external LD reference panel. SNP effect sizes were simulated using (1) a point-normal model as specified in Eq. (12) with different numbers of causal variants (100, 1000, 10,000, and 100,000), which represent extremely sparse to highly polygenic genetic architectures; and (2) a normal mixture model comprised 10 group-one SNPs, 1000 group-two SNPs and 10,000 group-three SNPs, and the three effect size groups

explained 10%, 20%, and 70% of the total heritability, respectively. The simulated trait was generated by the sum of all genetic markers, weighted by their simulated effect sizes, and adding a normally distributed noise term which fixed the heritability at 0.5. We then conducted GWAS to produce a marginal least squares effect size estimate for each SNP, and applied each polygenic prediction method to the GWAS summary statistics. For P+T, LDpred, and PRS-CS, tuning parameters were selected in a validation data set of 3000 individuals that are unrelated to the training sample. The predictive performance of all the six methods was evaluated in 3000 individuals (the testing set) that are unrelated to both the training sample and the validation set. $R^2$ between the observed and predicted traits was used to quantify the prediction accuracy. We regressed the true phenotype onto the PRS predictor, and used the regression slope as a measure of calibration. A slope close to one indicates that a predictor is well calibrated. For each combination of the genetic architecture and the training sample size (10,000, 20,000, 50,000, and 100,000), the simulation was repeated 20 times.

In order to systematically compare polygenic prediction methods across a wide range of settings, we conducted a number of secondary simulation studies: (1) sampling SNP effect sizes using a point-normal model with heritability fixed at 0.2 or 0.8; (2) sampling SNP effect sizes using a point-$t$ model with heavy tails (a mixture of a point mass at zero and a Student's $t$-distribution with 4 degrees of freedom); (3) sampling SNP effect sizes using a point-gamma model (a mixture of a point mass at zero and a gamma distribution with the shape parameter set to 2), which produces an effect size distribution that is asymmetric about zero and positively skewed with the right tail being long and thin and the left tail being short and fat; (4) using the combined UK Biobank validation and testing data sets ($N = 6000$) as an in-sample LD reference panel in the point-normal simulations. For each setting and training sample size considered (10,000, 20,000, 50,000, and 100,000), and the simulation was repeated 20 times.

**Partners HealthCare Biobank genetic data**. The Partners HealthCare Biobank [https://biobank.partners.org] is a collection of plasma, serum, DNA and buffy coats samples collected from consented subjects, which are linked to their electronic health records (EHR) and survey data on lifestyle, environment, and family history[55]. To date, Partners Biobank has enrolled more than 96,000 participants, and released genome-wide genetic data for 25,482 subjects. A study protocol is not required for Partners investigators to obtain de-identified data sets from Partners Biobank.

We performed QC on each genotyping batch separately with the following steps: (1) SNPs with genotype missing rate >0.05 were removed; (2) samples with genotype missing rate >0.02 or absolute value of heterozygosity >0.2, or samples that failed sex checks were excluded; (3) SNPs with missing rate >0.02, or HWE test $P < 1 \times 10^{-6}$ were discarded. We then removed SNPs that showed significant batch associations with $P < 1 \times 10^{-6}$, and merged genotyping batches for subsequent processing and analyses.

The Partners HealthCare Biobank included individuals from diverse populations. We used the 1KG samples as a population reference panel to infer the ancestry of Partners Biobank participants. Specifically, we computed principal components (PCs) of the genotype data in all the 1KG samples, and trained a random forest model using the top 4 PCs on the super population labels (African [AFR], American [AMR], East Asian [EAS], European [EUR], and South Asian [SAS]), in which EUR ($N = 503$) included TSI, IBS, GBR, CEU, and FIN subpopulations. The random forest model was then applied to the Partners Biobank participants, and identified 19,136 unrelated subjects ($\hat{\pi} < 0.2$) with European ancestry.

We used the Eagle2 software[56] [https://data.broadinstitute.org/alkesgroup/Eagle] for pre-phasing and Minimac3[57] [https://genome.sph.umich.edu/wiki/Minimac3] for imputation in the Partners Biobank European sample. Lastly, we removed markers with MAF <1%, missing rate >2%, imputation quality INFO score <0.8, and significant deviation from HWE with $P < 1 \times 10^{-10}$. All genetic analyses in the Partners Biobank were conducted using PLINK 1.9[54].

**Partners Biobank curated disease populations and quantitative traits**. For a number of common complex diseases, the Partners Biobank trained and validated a classification algorithm, which leverages both structured and unstructured EHR data, and combines natural language processing and statistical methods, in a gold standard training set created by expert chart review. The algorithm was then applied to all the participants in the Biobank to identify cases and controls, and create curated disease populations. We selected six curated diseases—BRCA, CAD, DEP, IBD (Crohn's disease or ulcerative colitis), RA, and T2DM—for which there are more than 500 cases in the Biobank that have been genotyped, and external large-scale GWAS summary statistics are publicly available. For all the diseases, cases have an algorithm-based positive predictive value (PPV) of having current or past history of the disease greater than 0.90, and controls have a negative predictive value (NPV) of having no history of the disease greater than 0.99.

In addition, we selected six quantitative traits—height (HGT), body mass index (BMI), high-density lipoproteins (HDL), low-density lipoproteins (LDL), cholesterol (CHOL), and triglycerides (TRIG)—that have been measured in the Partners Biobank healthy control population with a Charlson age-comorbidity index 0–2 and the predicted 10-year survival probability greater than 90%. We predicted these quantitative traits in a relatively heathy population to avoid

measurements affected by severe diseases or medications. For participants that have multiple measurements of a trait of interest, we used the median value. Table 1 presents the sample size for each curated disease and quantitative trait in the Partners Biobank.

**Summary statistics and polygenic prediction**. GWAS summary statistics for all the diseases and quantitative traits are publicly available (Supplementary Data 1). We removed ambiguous (A/T and C/G) SNPs and mapped the genetic markers to the Genome Reference Consortium human genome build 37. SNP heritability for each disease and trait was estimated using GWAS summary statistics and LD score regression[23]. Heritability estimates for diseases on the observed scale were transformed to the liability scale as described in Lee et al.[58] using the assumed population and sample prevalences shown in Supplementary Table 13. For unadjusted PRS and P+T, we used all the genetic markers that are present in the summary statistics, LD reference panel and the Partners Biobank genetic data. For LDpred (-inf) and PRS-CS(-auto), we further restricted the genetic markers to the HapMap3 panel to reduce memory and computational cost. Table 1 shows the total number of markers included in the analysis for each disease and quantitative phenotype. We note that the GWAS samples and the Partners Biobank sample may have overlap. However, by carefully examining the sample composition of each GWAS study, we believe that sample overlap is minimal (if any) and does not impact the comparison among polygenic prediction methods.

For each curated disease and quantitative trait, the Partners HealthCare Biobank sample was repeatedly and randomly split into a validation set comprising 1/3 of the data and a testing set comprising 2/3 of the data. Tuning parameters (P-value threshold in P+T, fraction of causal SNPs in LDpred, and global shrinkage parameter in PRS-CS) were selected in the validation set, and the predictive performance was evaluated in the testing set. We use the average $R^2$ between the observed and predicted phenotypes across 100 random splits to assess the predictive performance for the quantitative traits, and report the average Nagelkerke's $R^2$ metric across 100 random splits for disease (case–control) phenotypes. Nagelkerke's $R^2$ is defined as $R^2_{nag} = R^2/R^2_{max}$, where $R^2 = 1 - (\mathcal{L}_{res}/\mathcal{L}_{full})^{2/N}$, $R^2_{max} = 1 - \mathcal{L}^{2/N}_{res}$, $\mathcal{L}_{res}$ is the likelihood of a restricted logistic regression model with covariates only (an intercept, current age, sex and top 10 PCs of the genotype data), $\mathcal{L}_{full}$ is the likelihood of the full logistic regression model (covariates and the PRS predictor), and $N$ is the sample size. We define the relative increase or decrease in $R^2$ of a polygenic prediction method A compared to method B as $(R^2_A - R^2_B)/R^2_B$. In addition to $R^2$ or Nagelkerke's $R^2$, we also report area under the ROC curve (known as AUC), area under the precision-call curve, and the odds ratio (OR) comparing top 10% of the participants having high polygenic risk with the remaining 90% of the sample. We adjusted for current age, sex and top 10 PCs of the genotype data in the calculation of all predictive performance metrics.

**Reporting summary**. Further information on research design is available in the Nature Research Reporting Summary linked to this article.

## Data availability

UK Biobank data are available to registered investigators under approved applications [http://www.ukbiobank.ac.uk]. All genome-wide association summary statistics used in this study are publicly available. Download links are included in Supplementary Data 1. Other relevant data are available from the corresponding author upon request.

## Code availability

A Python package for PRS-CS is available on github repository [https://github.com/getian107/PRScs].

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

## Acknowledgements

This work involved the use of the Enterprise Research Infrastructure & Services (ERIS) at Partners HealthCare. We thank the Partners HealthCare Biobank for providing genomic and health information data. This research was funded in part by National Institutes of Health (NIH) U01HG008685 supporting the eMERGE Network, and K99AG054573 (T. G.). J.W.S. is a Tepper Family MGH Research Scholar and was supported in part by a gift from the Demarest Lloyd, Jr. Foundation. The funders had no role in study design, data collection and analysis, decision to publish, or preparation of the manuscript. This research has been conducted using the UK Biobank resource under an approved data request (ref: 32568). The breast cancer genome-wide association analyses were supported by the Government of Canada through Genome Canada and the Canadian Institutes of Health Research, the 'Ministère de l'Économie, de la Science et de l'Innovation du Québec' through Genome Québec and grant PSR-SIIRI-701, The National Institutes of Health (U19CA148065, X01HG007492), Cancer Research UK (C1287/A10118, C1287/A16563, C1287/A10710) and The European Union (HEALTH-F2-2009-223175 and H2020 633784 and 634935). All studies and funders are listed in Michailidou et al.[59]. Data on coronary artery disease have been contributed by CARDIoGRAMplusC4D investigators and have been downloaded from http://www.cardiogramplusc4d.org.

## Author contributions

T.G. conceived the study. T.G. and C.-Y.C. designed the experiments. T.G. developed the statistical methods with contributions from Y.N. C.-Y.C. preprocessed the Partners HealthCare Biobank genetic data. T.G. performed the simulations and real data analyses, with contributions from C.-Y.C. and Y.-C.A.F. T.G. developed the software, with input from C.-Y.C. and Y.-C.A.F. T.G. wrote the paper. C.-Y.C., Y.N., Y.-C.A.F., and J.W.S. provided critical revision for the manuscript. All authors reviewed and approved the final version of the manuscript.

## Additional information

**Competing interests:** The authors declare no competing interests.

