## [Peer Review File · Nature Communications]

Reviewer #1 (Remarks to the Author):

Polygenic Prediction via Bayesian Regression and Continuous Shrinkage Priors

Tian Ge et al.

Ge and colleagues introduced a novel genomic prediction method, polygenic risk scores with a continuous shrinkage (PRS-CS), using GWAS summary statistics and an external linkage disequilibrium panel. With simulations, the authors demonstrated that PRS-CS outperformed existing methods (standard PRS with and without LD-pruning and p-value threshold cut-off, LDpred and so on). The authors also tested the proposed method for a real data set.

The paper reads well. It is intriguing that the continuous shrinkage approach significantly improves the performance. I think the proposed approach is likely to have a large impact on the field. However, I have a number of comments that should be carefully considered to make sure that the main conclusion holds in general.

The proposed approach used an external LD panel (e.g. 1KG). In the analyses (for simulated as well as real data), the authors used imputed data and the imputations were somehow associated with 1KG data. Were there any confounding effects due to such artefact association such that the performance of the methods was influenced? A recent study showed that if LD scores were estimated from external reference samples and there was heterogeneity between the external reference and in-sample, estimates of genetic variances based on GWAS summary stats could be biased (AJHG 102: 1185-1194 (2018)).

1. I would suggest using raw genotypes in the target data set to see if there is any difference in their performance for the real data analyses.
2. How is the ranking of the methods changed if LD scores were estimated from the in-sample reference data (i.e. UK Biobank) in the simulation?
3. Should the authors comment or discuss about using a misspecified LD panel? For example, there is uncertainty about the target sample, i.e. prediction in forensic.

SEP

It is a known problem for PRS that the scale of the predictor (x) and response (y) is largely different, i.e. regression coefficient of y on x is far from 1. This is particularly important in a clinical practice that requires standardized risk profile scores (unbiased scale) across different cohorts.

4. Does PRS-CS have a better property in the unbiasedness than other existing methods?

Given parameters (Supplementary Table 3), one can quantify what would be an expected accuracy from theory (e.g. PLOS ONE 12, e0189775 (2017) and related software web page is

<https://sites.google.com/site/honglee0707/mtg2> (see section 9)). For height, assuming $h^2 = 0.5$, sample size in the reference = 693529, # SNPs = 2.3M, and effective number of chromosome segment (effectively independent number of SNPs) = 50,000, the expected prediction accuracy would be $R^2 = 0.41$ (quite different from that in Figure 2). I assume that the authors did a relatedness cut-off QC (0.025) such that effective number of chromosome segment was as large as 50,000 (see PLoS Genetics 10, e1004269 (2014)). Otherwise, the expected accuracy could be even higher. For BMI, the expected accuracy would be 0.18, assuming $h^2 = 0.25$.

5. Should the authors check where this difference came from?

6. It would be informative and clearer if the authors provide SNP-heritability estimation for the data in supplementary table 3.

7. I might miss. But it was not very clear how the authors dealt with high relationships in the simulation as well as real data analyses? It was mentioned that 488,377 samples were used for the UK Biobank, implying that the analyses included high relationships. It was not clear how these high relationships affected the performance of the methods.

8. Did the authors check if there were no overlapping samples between the reference data and Partners HealthCare Biobank genetic data?

A recent study reported that the top 0.5% of the participants according to their PRS were at fivefold increased risk for CAD, compared to the general population (Nature Genetics 50: 1219–1224 (2018)). Because Nagelkerke R^2 is hard to interpret in the aspect of clinical care, It would make more sense to report odds ratio (OR) of case-control status contrasting top 1% (or any %) with the general population.

9. Would it be possible to report OR from PRS-CS?

10. Figure 2 may need standard error bars.

It is not clear if the authors thoroughly checked the robustness of misspecified continuous mixing density.

11. Should the authors simulate SNP effects from a highly skewed gamma distribution to see if PRS-CS still performs the best?

I sign my name. S. Hong Lee.

Reviewer #2 (Remarks to the Author):

This is an interesting and useful paper. However, I think it overstates the case for this being a major advance. Or, to put it another way, I think you could explain the background and the benefits of the new method more clearly.

The introduction focuses on methods that use summary data as does this paper. However, it would be better to point out at the beginning that these are all approximations for methods described that use individual level data. And in fact all these methods derive from those described in your reference 34. The idea of using an LD reference to approximate the analysis was introduced by Yang et al (2012) Nature genetics 44;369.

You could make it clearer why you prefer a continuous shrinkage prior over one with a point mass at zero. The CS prior allows you to update the SNP effect estimates using BLUP. This is efficient in that you do a block update. However, this is only possible because you process the data in small genome blocks and assume no LD between the blocks. This means that you don't have to use "iteration on the data" because within a small block you can form the snp x snp equations. Many methods would be efficient under these circumstances. The reason other methods take a lot of computer time is that they fit so many SNPs simultaneously (allowing for LD amongst them all) that they cannot form the snp x snp equations and must use iteration on the data.

Incidentally, reference 34 includes a CS prior in the method called Bayes A.

Therefore the claim PRS-CS outperforms all other methods is a little too strong. For instance, in comparisons, bayes B or bayes R perform as well or better than all others. These could be turned into summary statistic based method just as has been done for PRS-CS.

Reviewer #3 (Remarks to the Author):

This is a well-written manuscript that proposes a new approach for polygenic risk scoring, based on a continuous rather than a discrete mixture prior. The method is clear and seems to be highly effective. I have some concerns mentioned below, mostly related to further elaborating the method's behavior under various scenarios.

Major concerns

1. The authors mention in various places that their method is computationally efficient, but they never provide any evidence. Please add runtime measurements and explain the factors that affect computational runtime. For example, each iteration requires inverting an $M \times M$ matrix where M is the block size, so I expect that the method would be much slower if you didn't filter to only HM3 SNPs.

2. In addition to (or instead of) reporting Nagelkerke's R^2 for disease PRS, please report either the area under the ROC curve or (better yet) the area under the precision-recall curve, as it informs of the actual predictive capability. Furthermore, the Methods section says that the phenotypes were adjusted for age, sex and PCs, so it's not clear to me how you could use Nagelkerke's R^2 , as it assumes a binary outcome.

3. All the simulations were conducted with $h^2=0.5$, which is higher than what we typically see in practice. I think a few more h^2 values should be evaluated.

4. The use of a reference sample size of only ~ 500 individuals (instead of the data from 500K individuals) is obviously a rough approximation. A recent paper[1] argues that the approximation becomes increasingly worse as the sample size increases (in the context of fine-mapping, but this should follow the same principles). Can the authors evaluate the sensitivity to the reference sample size? This can be done in simulations, by computing in-sample LD based on subsets of individuals from the UK Biobank (that are different from those used in the actual experiments).

Minor concerns

- "LDPred does not adequately adjust for the LD structure": Can you explain what this means? Is this a numerical problem (e.g. inversion of a near-singular matrix), or a conceptual one? I see that this is further discussed in the first discussion paragraph, but it's still not clear what is the approximation mentioned there.

- The discussion mentions explained variance of disease. This doesn't make sense, since the variance of a binary variable is a function of its mean. I suppose the authors think about variance explained on the liability scale, but then one would need to apply a correction factor that depends on the trait prevalence (e.g. [2]) to obtain a meaningful comparison.

- The author mention that they typically explain only a small portion of trait heritability. A recent paper claims that it managed to explain all of the heritability of human height via Lasso[3]. How do the authors explain this discrepancy?

- The authors used a different number of MCMC iterations (and burn-in iterations) in the simulations and the real data results. Can they provide some guidelines about choosing these parameters?

- Top of page 9: "The relative improvement in prediction accuracy": Please specify the exact accuracy measure you used

Even more minor concerns

- Page 8 above Table 1: "for which external large-scale GWAS summary statistics are publicly available": I found this sentence confusing at first, because I thought this described PBK in general.

- I would replace the "&" in the headers of Table 1 with a set-intersection sign?

- Table 1: I assume HM3 is Hapmap3? This isn't mentioned in the caption

- Table 1 caption should state what was the MAF cutoff.

- Why is ref. 59 (from 2009) used for Minimac3?

[1] Benner, Christian, et al. "Prospects of fine-mapping trait-associated genomic regions by using summary statistics from genome-wide association studies." *The American Journal of Human Genetics* 101.4 (2017): 539-551.

[2] Lee, Sang Hong, et al. "Estimating missing heritability for disease from genome-wide association studies." *The American Journal of Human Genetics* 88.3 (2011): 294-305.

[3] Lello, Louis, et al. "Accurate genomic prediction of human height." *Genetics* 210.2 (2018): 477-497.

NCOMMS-18-35083-T

Polygenic Prediction via Bayesian Regression and Continuous Shrinkage Priors

Tian Ge, Chia-Yen Chen, Yang Ni, Yen-Chen Anne Feng, Jordan W. Smoller

We were pleased to see a high level of enthusiasm of our work. We thank the three reviewers for their close read of the manuscript, insightful comments and helpful suggestions. We have carefully considered the points made by the reviewers and revised the manuscript accordingly. Major changes to the text are highlighted in red in the updated manuscript.

Below, we provide a point-by-point response to specific reviewer concerns. Our response is preceded by a double-dash (--) and rendered in a blue typeface. We believe that the manuscript is improved and hope it is now suitable for publication.

Reviewer #1 (Remarks to the Author):

Polygenic Prediction via Bayesian Regression and Continuous Shrinkage Priors

Tian Ge et al.

Ge and colleagues introduced a novel genomic prediction method, polygenic risk scores with a continuous shrinkage (PRS-CS), using GWAS summary statistics and an external linkage disequilibrium panel. With simulations, the authors demonstrated that PRS-CS outperformed existing methods (standard PRS with and without LD-pruning and p-value threshold cut-off, LDpred and so on). The authors also tested the proposed method for a real data set.

The paper reads well. It is intriguing that the continuous shrinkage approach significantly improves the performance. I think the proposed approach is likely to have a large impact on the field. However, I have a number of comments that should be carefully considered to make sure that the main conclusion holds in general.

The proposed approach used an external LD panel (e.g. 1KG). In the analyses (for simulated as well as real data), the authors used imputed data and the imputations were somehow associated with 1KG data. Were there any confounding effects due to such artefact association such that the performance of the methods was influenced? A recent study showed that if LD scores were estimated from external reference samples and there was heterogeneity between the external reference and in-sample, estimates of genetic variances based on GWAS summary stats could be biased (AJHG 102: 1185-1194 (2018)).

1. I would suggest using raw genotypes in the target data set to see if there is any difference in their performance for the real data analyses.

-- Thanks for the suggestion. We have repeated the real data analyses using raw genotypes in the Partners HealthCare Biobank. The prediction accuracy (Nagelkerke R^2 for disease phenotypes and R^2 for quantitative

traits) for each polygenic prediction method is summarized in the table below. Using raw genotypes resulted in fewer SNPs being included in prediction (as shown in the “# SNPs” column), and thus the prediction accuracy was in general slightly lower than using the imputed data. However, we note that the ranking of the polygenic prediction methods did not change. In addition, in the revised manuscript, we have also included the predictive performance of all methods when an in-sample LD reference panel is used in both simulation studies (see Supplementary Figure 5 and Supplementary Table 6) and real data analyses (see Supplementary Figure 6 and Supplementary Table 16). These results were highly consistent to those using the external 1KG reference panel. Combining these results, it is unlikely that the improvement of PRS-CS over alternative methods is due to the artefact association induced by using the same reference panel for imputing the target sample and inferring posterior SNP effect sizes.

Disease/Trait	# SNPs	PRS-CS	LDpred	P+T	Ldpred-inf	PRS-unadj
Breast Cancer	395,250	0.0433	0.0206	0.0255	0.0233	0.0177
Coronary Artery Disease	389,572	0.0341	0.0325	0.0168	0.0190	0.0217
Depression	392,103	0.0087	0.0092	0.0070	0.0072	0.0075
Inflammatory Bowel Disease	391,616	0.0663	0.0523	0.0495	0.0262	0.0299
Rheumatoid Arthritis	349,908	0.0783	0.0630	0.0614	0.0633	0.0349
Type 2 Diabetes Mellitus	394,452	0.0437	0.0402	0.0206	0.0220	0.0235
Height	259,327	0.2639	0.2234	0.2223	0.2403	0.1794
Body mass index	259,272	0.1035	0.0921	0.0827	0.0927	0.0587
High-density lipoproteins	266,743	0.0885	0.0904	0.0488	0.0449	0.0265
Low-density lipoproteins	266,187	0.0972	0.0842	0.0788	0.0372	0.0277
Cholesterol	266,723	0.0785	0.0580	0.0599	0.0379	0.0267
Triglycerides	266,251	0.0579	0.0518	0.0386	0.0293	0.0210

2. How is the ranking of the methods changed if LD scores were estimated from the in-sample reference data (i.e. UK Biobank) in the simulation?

-- As mentioned above, we have included the predictive performance of all polygenic prediction methods considered when an in-sample LD reference panel is used in both simulation studies (see Supplementary Figure 5 and Supplementary Table 6) and real data analyses (see Supplementary Figure 6 and Supplementary Table 16). Using a larger in-sample LD reference panel in general produced slightly higher prediction accuracy. However, the improvement was marginal, and it appears that the predictive performance of PRS-CS is not particularly sensitive to the selection of LD reference panel when the GWAS, reference and target samples have matched ancestry.

3. Should the authors comment or discuss about using a misspecified LD panel? For example, there is uncertainty about the target sample, i.e. prediction in forensic.

-- We have now included a comment on misspecified LD panels along with cross-ethnic risk prediction in the Discussion section (see Pages 13-14).

It is a known problem for PRS that the scale of the predictor (x) and response (y) is largely different, i.e. regression coefficient of y on x is far from 1. This is particularly important in a clinical practice that requires standardized risk profile scores (unbiased scale) across different cohorts.

4. Does PRS-CS have a better property in the unbiasedness than other existing methods?

-- Thanks for raising this important question. We have regressed the true phenotype onto the PRS predictor and reported the regression slope as a measure of calibration for each simulation setting in the revised manuscript (see Supplementary Tables 7-12). Consistent with predictive performance, as the training sample size grows, our Bayesian approach provides the best calibration among all methods examined. PRS-CS-auto is particularly well calibrated for large training sample sizes because it automatically learns the sparseness of the genetic architecture and adjusts for the LD structure accordingly.

Given parameters (Supplementary Table 3), one can quantify what would be an expected accuracy from theory (e.g. PLOS ONE 12, e0189775 (2017) and related software web page is <https://sites.google.com/site/honglee0707/mtg2> (see section 9)). For height, assuming $h^2 = 0.5$, sample size in the reference = 693529, # SNPs = 2.3M, and effective number of chromosome segment (effectively independent number of SNPs) = 50,000, the expected prediction accuracy would be $R^2 = 0.41$ (quite different from that in Figure 2). I assume that the authors did a relatedness cut-off QC (0.025) such that effective number of chromosome segment was as large as 50,000 (see PLoS Genetics 10, e1004269 (2014)). Otherwise, the expected accuracy could be even higher. For BMI, the expected accuracy would be 0.18, assuming $h^2 = 0.25$.

5. Should the authors check where this difference came from?

-- A few reasons can result in this discrepancy. First, as we now report in Supplementary Table 14, the SNP heritability of height was estimated to be 0.45 using the GIANT GWAS summary statistics and LD score regression, which is slightly lower than 0.5. Second, previous theoretical results of prediction accuracy were derived based on infinitesimal models with a normal prior. Therefore, they may not provide close approximations to non-infinitesimal models or more sophisticated prior distributions. Third, we did not use all SNPs in GWAS summary statistics in prediction, but restricted the analysis to the 1KG reference panel and HapMap3 SNPs. Therefore, the total number of SNPs included in height prediction was ~750K as reported in Table 1 instead of 2.3M. This may affect the effective number of independent SNPs as well. We have discussed the rationale and limitations of restricting the analysis to HapMap3 SNPs in the revised manuscript, and pointed out that including multi-million SNPs predictors may increase prediction accuracy but requires future work, referencing the PLoS ONE article (see Pages 12-13).

6. It would be informative and clearer if the authors provide SNP-heritability estimation for the data in supplementary table 3.

-- Thanks for the suggestion. We have now provided SNP heritability for each disease (both on the observed scale and on the liability scale) and trait, estimated using GWAS summary statistics and LD score regression, in Supplementary Table 14 for reference.

7. I might miss. But it was not very clear how the authors dealt with high relationships in the simulation as well as real data analyses? It was mentioned that 488,377 samples were used for the UK Biobank, implying that the analyses included high relationships. It was not clear how these high relationships affected the performance of the methods.

-- Sorry for the confusion. 488,377 is the total sample size for the UK Biobank. In our simulation studies, we restricted the analysis to unrelated white British participants along with other sample QC procedures (see the Methods section on Page 20), and randomly selected 10K, 20K, 50K or 100K subjects as the GWAS (training) sample (see the description of simulation design on Page 20). The validation and testing data sets also included unrelated participants and are independent from the training set. Therefore, the performance of polygenic prediction methods is not affected by sample relatedness in the UK Biobank. Similarly, in real data analyses, we selected unrelated subjects with European ancestry (see Page 21).

8. Did the authors check if there were no overlapping samples between the reference data and Partners HealthCare Biobank genetic data?

-- There is no sample overlap between the 1KG European sample and the Partners HealthCare Biobank sample. We cannot exclude the possibility of sample overlap between each GWAS sample and the Partners Biobank sample. However, by carefully examining the sample composition of each GWAS study, we believe that sample overlap is minimal (if any) and does not impact the comparison among polygenic prediction methods.

A recent study reported that the top 0.5% of the participants according to their PRS were at fivefold increased risk for CAD, compared to the general population (Nature Genetics 50: 1219–1224 (2018)). Because Nagelkerke R^2 is hard to interpret in the aspect of clinical care, it would make more sense to report odds ratio (OR) of case-control status contrasting top 1% (or any %) with the general population.

9. Would it be possible to report OR from PRS-CS?

-- Yes. PRS-CS can produce any predictive performance metric that other polygenic prediction methods can provide. However, since the sample size of the Partners HealthCare Biobank is much smaller than the UK Biobank, we do not have enough cases to reliably calculate OR for the extreme tail of the PRS distribution (e.g., top 1% or lower). In the revised manuscript, we report OR contrasting top 10% of the participants having high polygenic risk with the remaining 90% of the sample as an example. PRS-CS ORs range from 1.58 for depression to 4.38 for rheumatoid arthritis, which are higher than alternative methods and are consistent with prediction accuracy. As suggested by another reviewer, we also report area under the receiver operating characteristic (ROC) curve (AUC) and area under the precision-recall curve in the updated manuscript, both are clinically relevant (see Supplementary Tables 15 and 16). All prediction accuracy metrics produced consistent results in terms of the ranked performance of polygenic prediction methods.

10. Figure 2 may need standard error bars.

-- We have added the standard deviation of the prediction accuracy (R^2 or Nagelkerke R^2) for each trait or disease across 100 random splits of the Partners HealthCare Biobank sample to the figure.

It is not clear if the authors thoroughly checked the robustness of misspecified continuous mixing density.

11. Should the authors simulate SNP effects from a highly skewed gamma distribution to see if PRS-CS still performs the best?

-- The gamma-gamma prior on the local shrinkage parameter is designed to create a prior distribution on the SNP effect sizes, which has a sizable amount of mass near zero to impose strong shrinkage on noise and at the same time has heavy tails to avoid over-shrinkage of truly non-zero effects. Therefore, the shape of the prior on effect sizes (peaky at zero and heavy-tailed) is important while the exact scale-mixture distribution that generates this shape is less important. In fact, we had simulated SNP effect sizes from a point-normal model, a normal mixture model and a point-t model, all of which are different from the gamma-gamma model and thus the impact of misspecified effect size distributions had been investigated. In this revised version of the manuscript, we added one more simulation, which sampled SNP effect sizes from a point-gamma model (a point mass at zero and a gamma distribution with the shape parameter set to 2), which produces an effect size distribution that is asymmetric about zero and highly skewed with the right tail being long and thin and the left tail being short and fat. We show that the performance of PRS-CS is not sensitive to specific effect size distributions, and consistently improves over alternative methods in all simulation settings (see Figure 1 and Supplementary Figures 1-4).

Reviewer #2 (Remarks to the Author):

This is an interesting and useful paper. However, I think it overstates the case for this being a major advance. Or, to put it another way, I think you could explain the background and the benefits of the new method more clearly.

The introduction focuses on methods that use summary data as does this paper. However, it would be better to point out at the beginning that these are all approximations for methods described that use individual level data. And in fact, all these methods derive from those described in your reference 34. The idea of using an LD reference to approximate the analysis was introduced by Yang et al (2012) Nature genetics 44;369.

-- In the revised version of the manuscript, we have pointed out, when laying out the conceptual framework of polygenic prediction methods, that individual-level models can often be approximated using an external LD reference panel and turned into summary statistics based methods, and have cited Yang et al. (2012) and the very recent preprint by Lloyd-Jones et al. which extended BayesR to a summary statistics based method. See Page 5.

You could make it clearer why you prefer a continuous shrinkage prior over one with a point mass at zero. The CS prior allows you to update the SNP effect estimates using BLUP. This is efficient in that you do a block update. However, this is only possible because you process the data in small genome blocks and assume no LD between the blocks. This means that you don't have to use "iteration on the data" because within a small block you can form the snp x snp equations. Many methods would be efficient under these circumstances. The reason other methods take a lot of computer time is that they fit so many SNPs simultaneously (allowing for LD amongst them all) that they cannot form the snp x snp equations and must use iteration on the data.

-- Thanks for providing this insight. We have discussed the advantages and limitations of using a genomic partition in the revised manuscript (see Pages 12-13). Specifically, we have made the following points: (1) Partitioning the genome into small LD blocks makes block update of continuous shrinkage priors feasible, but assumes that there is no LD between blocks. Using a sliding window approach as implemented in LDpred may more accurately capture LD across blocks, but is more memory and computationally expensive. (2) The computational cost of the algorithm depends on the size of the LD blocks: Expanding the size of LD blocks may improve prediction accuracy but increases computational cost (as each MCMC iteration requires inverting a snp x snp matrix), while reducing the size of LD blocks has the potential risk of missing long-range LD. Therefore, there is a balance between modeling accuracy and computational burden.

Incidentally, reference 34 includes a CS prior in the method called Bayes A.

-- We have cited Meuwissen et al. (2001) along with other prior work that applied special cases of continuous shrinkage priors (e.g., Bayesian LASSO) to genomic prediction in the Introduction (see Page 4). However, we also note that (1) all these work required individual-level data, although technically they can be turned into summary statistics based methods. (2) The continuous shrinkage prior we used in our method is carefully designed to create a prior distribution on the SNP effect sizes, which has a sizable amount of mass near zero to impose strong shrinkage on noise and at the same time has heavy tails to avoid over-shrinkage of truly non-zero effects. Therefore, it is robust to varying genetic architectures and offers much better shrinkage properties than many simple continuous shrinkage priors. For example, BayesA is essentially an infinitesimal model with a Student's t prior on SNP effect sizes, and thus does not impose strong shrinkage on small noisy estimates and would be difficult to handle sparse genetic architectures.

Therefore the claim PRS-CS outperforms all other methods is a little too strong. For instance, in comparisons, bayes B or bayes R perform as well or better than all others. These could be turned into summary statistic based method just as has been done for PRS-CS.

-- We agree that in principle all genomic prediction methods developed for individual-level data can be turned into summary statistics based methods and have made this explicit in the Conceptual Frameworks section (see Page 5). We have also reviewed genomic prediction methods for individual-level data in the Discussion section (see Page 13). However, given that a majority of the existing genomic prediction models use discrete mixture priors, we believe that continuous shrinkage priors provide an alternative perspective in polygenic prediction, and have advantages in computational efficiency and multivariate LD modeling (through block update of SNP effect sizes) and flexible modeling of genetic architectures (through the design of the shape of priors on SNP

effect sizes). By better referencing prior work and clarifying the motivation of using continuous shrinkage priors as discussed above, we hope that we have more precisely conveyed the contribution of this work to the community in this revised version of the manuscript.

Reviewer #3 (Remarks to the Author):

This is a well-written manuscript that proposes a new approach for polygenic risk scoring, based on a continuous rather than a discrete mixture prior. The method is clear and seems to be highly effective. I have some concerns mentioned below, mostly related to further elaborating the method's behavior under various scenarios.

Major concerns

1. The authors mention in various places that their method is computationally efficient, but they never provide any evidence. Please add runtime measurements and explain the factors that affect computational runtime. For example, each iteration requires inverting an $M \times M$ matrix where M is the block size, so I expect that the method would be much slower if you didn't filter to only HM3 SNPs.

-- We have now added runtime measurements in the Discussion section of the manuscript (see Pages 12-13). Specifically, we fit a regression model for each chromosome and the Bayesian computation with 1,000 MCMC iterations on the longest chromosome can be completed within an hour using one Intel(R) Xeon(R) CPU core and 2GB of memory. Therefore, all computation (with 1,000 MCMC iterations) can be finished in an hour if the 22 chromosomes are processed in parallel. Computational runtime is difficult to fairly compare between Bayesian methods. When we say our method is "computationally efficient", we refer more to the better mixing and convergence properties of continuous shrinkage priors relative to discrete mixture priors, which have been well studied and documented in the statistics literature. We have made this clearer in the revised manuscript in various places to avoid confusion.

We have also discussed factors that affect computational burden. The genomic partition we used gives a moderate number of HapMap3 SNPs within each LD block (on average ~ 500 SNPs per block), which makes iterative matrix inversion feasible. Expanding the size of LD blocks may improve prediction accuracy but increases computational cost, while reducing the size of LD blocks has the potential risk of missing long-range LD. Therefore, our choice of this particular genomic partition represents a balance between modeling accuracy and computational burden. The reviewer is totally correct that if we do not restrict the analysis to HapMap3 SNPs, the computational cost will increase dramatically. We have discussed this in the revision, and pointed out that further work is needed to include multi-million SNP predictors in the prediction (see Pages 12-13).

2. In addition to (or instead of) reporting Nagelkerke's R^2 for disease PRS, please report either the area under the ROC curve or (better yet) the area under the precision-recall curve, as it informs of the actual predictive capability. Furthermore, the Methods section says that the phenotypes were adjusted for age, sex and PCs, so

it's not clear to me how you could use Nagelkerke's R^2 , as it assumes a binary outcome.

-- We have added area under the receiver operating characteristic (ROC) curve (AUC) and area under the precision-recall curve for each disease to the revised manuscript. As suggested by another reviewer, we also report the odds ratio contrasting top 10% of the participants having high polygenic risk with the remaining 90% of the sample (see Supplementary Tables 15 and 16). All prediction accuracy metrics produced consistent results in terms of the ranked performance of polygenic prediction methods.

Nagelkerke's R^2 is defined as $R_{\text{nag}}^2 = R^2 / R_{\text{max}}^2$, where $R^2 = 1 - (\mathcal{L}_{\text{res}} / \mathcal{L}_{\text{full}})^{2/N}$, $R_{\text{max}}^2 = 1 - \mathcal{L}_{\text{res}}^{2/N}$, \mathcal{L}_{res} is the likelihood of a restricted logistic regression model with covariates only (an intercept, age, sex and top 10 PCs), $\mathcal{L}_{\text{full}}$ is the likelihood of the full logistic model (covariates and the PRS predictor), and N is the sample size. We have clarified this definition in the Methods section of the manuscript (see Pages 22-23).

3. All the simulations were conducted with $h^2=0.5$, which is higher than what we typically see in practice. I think a few more h^2 values should be evaluated.

-- We have added simulations of the point-normal model with $h^2=0.2$ and $h^2=0.8$ to the revised manuscript. The results (i.e., the ranking of predictive performance across polygenic prediction methods examined) are highly consistent with $h^2=0.5$ (see Supplementary Figures 1 and 2; Supplementary Tables 2 and 3).

4. The use of a reference sample size of only ~500 individuals (instead of the data from 500K individuals) is obviously a rough approximation. A recent paper [1] argues that the approximation becomes increasingly worse as the sample size increases (in the context of fine-mapping, but this should follow the same principles). Can the authors evaluate the sensitivity to the reference sample size? This can be done in simulations, by computing in-sample LD based on subsets of individuals from the UK Biobank (that are different from those used in the actual experiments).

-- In this revised version of the manuscript, we have included the predictive performance of all methods when an in-sample LD reference panel is used in both simulation studies (see Supplementary Figure 5 and Supplementary Table 6) and real data analyses (see Supplementary Figure 6 and Supplementary Table 16). In the simulation studies, the combined validation and testing data sets were used as an in-sample reference panel ($N=6,000$). In real data analyses, the Partners HealthCare Biobank sample was used as an in-sample reference panel ($N=19,136$). In general, the prediction accuracy was slightly increased when a larger in-sample LD reference was used. However, the improvement was marginal in both simulations and real data analyses. It appears that the performance of PRS-CS(-auto) is not particularly sensitive to the LD reference panel, and 1KG can serve as a valid reference despite its relatively small sample size. We have included these points in the updated manuscript (see Page 7). In addition, we have also briefly discussed the potential impact of a misspecified LD reference in the Discussion (see Pages 13-14).

Minor concerns

- "LDpred does not adequately adjust for the LD structure": Can you explain what this means? Is this a

numerical problem (e.g. inversion of a near-singular matrix), or a conceptual one? I see that this is further discussed in the first discussion paragraph, but it's still not clear what is the approximation mentioned there.

-- The approximation used by LDpred is quite technical and difficult to fully explain by text. The bottom line is that LDpred uses marginal posterior without LD to approximate the true multivariate posterior, and this approximation appears to be inaccurate. We have replaced "inadequate adjustment for the LD structure" with "inaccurate adjustment for the LD structure" in various places.

- The discussion mentions explained variance of disease. This doesn't make sense, since the variance of a binary variable is a function of its mean. I suppose the authors think about variance explained on the liability scale, but then one would need to apply a correction factor that depends on the trait prevalence (e.g. [2]) to obtain a meaningful comparison.

-- Thanks for pointing this out. We have rephrased the discussion in the last paragraph of the manuscript (see Page 14). Basically, what we would like to say here is that given the relatively low AUC and area under the precision-recall curve, the predictive performance of PRS needs to be further improved to increase its clinical value.

- The author mention that they typically explain only a small portion of trait heritability. A recent paper claims that it managed to explain all of the heritability of human height via Lasso [3]. How do the authors explain this discrepancy?

-- First, when we say PRS explains only a small portion of trait heritability, we referred to heritability estimated by twin studies. As suggested by another reviewer, we now provide SNP heritability for each disease and trait estimated using GWAS summary statistics and LD score regression in Supplementary Table 14. It can be seen that PRS predictors can explain a much larger portion of common-SNP heritability. Second, the prediction accuracy of human height reported in Lello et al. may be over-estimated for a couple of reasons: (1) The authors reported the maximum correlation between the true and predicted height in the testing set by varying the regularization parameter in LASSO, while in principle the regularization parameter should be estimated within the training set (e.g., via a folded cross-validation) and fixed when assessing predictive performance in the testing set. Therefore, the high prediction accuracy is likely to be a result of overfitting. (2) Both training and testing sets are UK Biobank participants, and LASSO was applied to individual-level data. Highly homogeneous training and testing data improves predictive performance, and overfitting the training data may increase prediction accuracy. In fact, as shown in their out-of-sample prediction example, when the LASSO predictor was trained on UK Biobank participants and applied to the Atherosclerosis Risk in Communities Study (ARIC), the correlation between the true and predicted height dropped to 0.536, which corresponds to an $R^2 < 0.3$ and is comparable to our results. We further note that frequentist LASSO has a Bayesian counterpart that can be represented as continuous shrinkage priors and posterior mode inferences (see discussion on Page 13). Therefore, our framework subsumes LASSO as a special case and enables the application of LASSO procedure to GWAS summary statistics.

- The authors used a different number of MCMC iterations (and burn-in iterations) in the simulations and the real data results. Can they provide some guidelines about choosing these parameters?

-- In general, using more MCMC iterations improves the mixing and convergence of the Markov chain. Therefore, we suggest running a longer Markov chain (e.g., 10,000 iterations) in practice when time and computational resources permit. However, we also found in simulation studies that the Gibbs sampler usually attains reasonable convergence after 1,000 MCMC iterations and produces prediction accuracy close to what can be achieved by much longer MCMC runs. Therefore, to reduce computational cost we used 1,000 iterations in simulations. We have made this clearer in the revised manuscript (see Page 18).

- Top of page 9: "The relative improvement in prediction accuracy": Please specify the exact accuracy measure you used.

-- We have clarified the definition of relative improvement in the Methods section (see Page 23). Specifically, the relative increase/decrease in R^2 of a polygenic prediction method A compared to B is defined as $(R_A^2 - R_B^2)/R_B^2$.

Even more minor concerns

- Page 8 above Table 1: "for which external large-scale GWAS summary statistics are publicly available": I found this sentence confusing at first, because I thought this described PBK in general.

-- We have split and rephrased this sentence as "Large-scale GWAS summary statistics for each disease and trait were downloaded from public domains (Table 1 and Supplementary Table 13)".

- I would replace the "&" in the headers of Table 1 with a set-intersection sign?

-- We have replaced "&" with a set-intersection sign in Table 1.

- Table 1: I assume HM3 is Hapmap3? This isn't mentioned in the caption.

-- Yes. We have spelled this out in the caption.

- Table 1 caption should state what was the MAF cutoff.

-- We have added MAF cutoff information in the caption.

- Why is ref. 59 (from 2009) used for Minimac3?

-- We have removed this incorrect reference. Thanks for catching this.

[1] Benner, Christian, et al. "Prospects of fine-mapping trait-associated genomic regions by using summary

statistics from genome-wide association studies." *The American Journal of Human Genetics* 101.4 (2017): 539-551.

[2] Lee, Sang Hong, et al. "Estimating missing heritability for disease from genome-wide association studies." *The American Journal of Human Genetics* 88.3 (2011): 294-305.

[3] Lello, Louis, et al. "Accurate genomic prediction of human height." *Genetics* 210.2 (2018): 477-497.

Reviewer #1 (Remarks to the Author):

The authors have addressed most of my concerns and the manuscript has been significantly improved.

I have a minor comment. The authors mentioned that they couldn't exclude the possibility of overlapping samples between the GWAS and the Partners Biobank data, which should be mentioned as a limitation of the study in Discussion.

Reviewer #3 (Remarks to the Author):

I thank the authors for their revision and their clarification. I am happy with the manuscript and have no further comments.

Signed: Omer Weissbrod

NCOMMS-18-35083A

Polygenic Prediction via Bayesian Regression and Continuous Shrinkage Priors

Tian Ge, Chia-Yen Chen, Yang Ni, Yen-Chen Anne Feng, Jordan W. Smoller

We thank all the reviewers for supporting the publication of this manuscript. Below, we provide a point-by-point response to outstanding issues raised by the reviewers. Our response is preceded by a double-dash (--) and rendered in a blue typeface.

Reviewer #1 (Remarks to the Author):

The authors have addressed most of my concerns and the manuscript has been significantly improved.

I have a minor comment. The authors mentioned that they couldn't exclude the possibility of overlapping samples between the GWAS and the Partners Biobank data, which should be mentioned as a limitation of the study in Discussion.

-- Thanks for the suggestion. We have discussed the potential impact of sample overlap between the GWAS samples and the Partners Biobank sample in the Methods section (see the first paragraph on Page 19).

Reviewer #3 (Remarks to the Author):

I thank the authors for their revision and their clarification. I am happy with the manuscript and have no further comments.